# Direct Imaging of the Kinetic Crystallization Pathway: Simulation and Liquid-Phase Transmission Electron Microscopy Observations

**DOI:** 10.3390/ma16052026

**Published:** 2023-03-01

**Authors:** Zhangying Xu, Zihao Ou

**Affiliations:** 1Qian Weichang College, Shanghai University, Shanghai 200444, China; 2Department of Materials Science and Engineering, Stanford University, Stanford, CA 94305, USA; 3Wu Tsai Neurosciences Institute, Stanford University, Stanford, CA 94305, USA

**Keywords:** crystallization, kinetics, direct imaging, liquid-phase transmission electron microscopy, computer simulations

## Abstract

The crystallization of materials from a suspension determines the structure and function of the final product, and numerous pieces of evidence have pointed out that the classical crystallization pathway may not capture the whole picture of the crystallization pathways. However, visualizing the initial nucleation and further growth of a crystal at the nanoscale has been challenging due to the difficulties of imaging individual atoms or nanoparticles during the crystallization process in solution. Recent progress in nanoscale microscopy had tackled this problem by monitoring the dynamic structural evolution of crystallization in a liquid environment. In this review, we summarized several crystallization pathways captured by the liquid-phase transmission electron microscopy technique and compared the observations with computer simulation. Apart from the classical nucleation pathway, we highlight three nonclassical pathways that are both observed in experiments and computer simulations: formation of an amorphous cluster below the critical nucleus size, nucleation of the crystalline phase from an amorphous intermediate, and transition between multiple crystalline structures before achieving the final product. Among these pathways, we also highlight the similarities and differences between the experimental results of the crystallization of single nanocrystals from atoms and the assembly of a colloidal superlattice from a large number of colloidal nanoparticles. By comparing the experimental results with computer simulations, we point out the importance of theory and simulation in developing a mechanistic approach to facilitate the understanding of the crystallization pathway in experimental systems. We also discuss the challenges and future perspectives for investigating the crystallization pathways at the nanoscale with the development of in situ nanoscale imaging techniques and potential applications to the understanding of biomineralization and protein self-assembly.

## 1. Introduction

Crystallization is a generic behavior that describes the emergence of an ordered structure from a parent disordered structure. As one of the most fundamental questions in materials synthesis and processing, crystallization has been extensively studied both theoretically and experimentally [1]. The properties of crystals, such as crystal structure, size distribution, domain size, and polymorphism, are strongly correlated with the crystallization pathways underlying their formation [2]. One of the central motivations for studying crystallization is to understand and engineer the underlying pathways of fundamental interest and crucially relevant to their functionalities, ranging from the solidification of rocks in the Earth’s mantle to the formation of bones through biomineralization [3]. Due to the stochastic nature of the initial nucleation of crystallization, directly capturing the motion of individual building blocks as they interact and crystallize would be effective to illustrate the kinetic transition of the system. However, although classical nucleation theory has been proposed to quantitatively predict the nucleation of the materials, the initial crystallization step has been blurred due to the difficulty of directly visualizing the early stage of crystal nuclei and tracking the growth of nanoparticles from the beginning to its crystalline form.

Crystallization theory is not only limited to the description of atoms forming into ordered atomic lattices, but it also applies to the self-assembly of a colloidal suspension into an ordered superlattice [4,5,6]. At the nanoscale, it has been pointed out that the interactions between individual nanoparticles or large molecules could be nonadditive, resulting in completely different self-assembly behaviors in solution, as well as much more complicated and richer varieties of phases in comparison to the well-studied model system of micron-sized colloidal systems [5]. In the past three decades, a variety of complex structures have been observed via self-assembly in solution [7], and recent examples include a quasicrystalline superlattice [8], clathrate crystals [9], and chiral pinwheel structures [10]. Understanding the crystallization pathways in these systems has posed new challenges in theory, and directly imaging the crystallization pathways in solution from individual nanoparticles at the nanoscale elucidates the interparticle interaction between these particles and facilitates the production of hierarchical materials for novel electronic [11] and high-performance photonic applications [12].

Advancements in in situ characterization methods have been beneficial to reveal the missing links in understanding crystallizations. In general, techniques for mapping the crystallization kinetics can be divided into two categories depending on whether they capture the motion of single particles in real space or display the collection of structural evolutions in the reciprocal space. For techniques that probe the reciprocal space, popular examples include X-ray spectroscopy [13] and optical light scattering [14]. Direct imaging techniques offer straightforward information about a material’s structure in contrast to the complicated data analysis required for reciprocal space observations. For direct imaging techniques, conventional optical imaging techniques are usually limited by the diffraction limit (~200 nm), and they are limited to the study of the crystallization kinetics of micron-sized colloidal particles [4,5] or visualize the crystal morphology without single particle resolution [15]. In situ techniques that can achieve a nanoscale resolution include super-resolution optical microscopy [16], scanning electron microscopy (SEM) [17], transmission electron microscopy (TEM) [18], and atomic force microscopy (AFM). [19] Among these techniques, liquid-phase TEM has been significantly developed in the past two decades [20], and this technique has been used in addition to other material characterization techniques to illustrate the dynamics of the materials in their native liquid environment [10]. By sandwiching the liquid droplet between two thin materials, this allows the observation of liquid samples in a high vacuum environment required by electron transport [21], and this technique has been utilized to study a variety of materials questions, including the nucleation and growth of two-dimensional (2D) materials [22], self-assembly kinetics [23], and other systems, such as geochemistry and electrochemistry [1,24]. It is noteworthy that ex situ techniques can also provide valuable information for the crystallization kinetics by revealing the structural details of the materials at a specific time during crystallization, and these demonstrations have been realized using cryo-electron microscopy [25], atomic electron tomography [26], etc. The crystallization process can also be studied in situ by monitoring order parameters that are not directly related to the structure of the materials, such as ellipsometry [27] and UV–Vis spectroscopy [28].

In this review, we focused on recent observations in liquid-phase TEM and compared side by side the observations with simulation results. First, we briefly discuss the crystallization pathways and data processing methods in liquid-phase TEM and computer simulations in Section 2. Experimental and simulation observations following classical and nonclassical crystallization pathways are discussed in Section 3 and Section 4, respectively. For the nonclassical crystallization pathways, three different pathways are discussed: (1) formation of an amorphous cluster below critical nucleus size; (2) nucleation of a crystalline phase from an amorphous intermediate; (3) transition between multiple crystalline structures before achieving the final product. In Section 5, we briefly discuss the postgrowth process of the crystal after the initial nucleation, as well as the challenges and perspectives on studying crystallization in liquid-phase TEM in Section 6.

## 2. Description and Analysis of Nucleation Pathways in Experiments and Simulation

### 2.1. Classical and Nonclassical Nucleation Pathways

In this review, we summarize four different crystallization pathways, as shown in Figure 1. All these pathways describe the formation from the original disordered atoms or particles to the final thermodynamically stable crystalline lattice structure. In classical nucleation theory (CNT), as shown in Figure 1a, the small nuclei follow the same crystalline structure as the final crystal, and the symmetry and density of the nuclei are maintained through the whole crystallization process. On the other hand, all other pathways involve intermediate structures that deviate from the classical crystallization pathway. In path 1 (Figure 1b), the nuclei are disordered below the critical nuclei size while changing to the final crystalline structure after growing beyond this limit. In path 2 (Figure 1c), the nuclei maintain a large, disordered structure, and small crystalline nuclei grow from the large, disordered phase. In path 3 (Figure 1d), the nuclei first evolve into a crystalline lattice with different symmetries, and the lattice transform to the other lattice at a later stage of nucleation. Although the existence of an intermediate state during crystallization has been predicted from density functional theory [29] and molecular dynamic simulations [30], the direct imaging of such intermediate structures in their native environments would benefit the validation and expansion of theory to encounter the increasing number of materials available to study, ranging from conventional materials, such as water [31] and salt [32], to novel 2D materials [22] and metal–organic frameworks [33].

One important message is that one system can have multiple crystallization pathways depending on the experimental or simulation conditions. For example, in the study of the crystallization of calcium carbonate, it has been pointed out that both the CNT and nonclassical crystallization pathways could be observed [34]. This is particularly clear in numerical simulations, where the initial states of the system can be accurately controlled. By changing the starting concentration and interaction strength, the systems could switch from CNT to nonclassical pathways, or both pathways could also exist at the same time [35]. We would like to also point out that the limited field of view in liquid-phase TEM may not capture the complete phase transition kinetics and techniques studying the structural evolution in the reciprocal space, such as in situ X-ray [13], which usually samples a much larger volume of the sample during crystallization, and they could be complimentary with liquid-phase TEM to reveal the full picture of the crystallization kinetics.

It is worth mentioning that the nonclassical pathways included in this review were selected based on the criteria that we can compare both simulation and experimental results from the literature. In many cases, the term nonclassical crystallization pathways are used, while the crystallization kinetics could be different from the CNT in any aspect, as pointed out in a seminal review on nonclassical crystallization pathways [2]. Going beyond the pathways summarized in Figure 1, other nonclassical crystallization pathways that differ from the CNT have also been reported. For example, different from the assumption in CNT that nuclei are always spherical, at the early stage of atomic nucleation, the shapes of the early-stage nuclei could be anisotropic, and the boundary between the crystalline core and the outside could be diffusive [26]. In another example of the self-assembly of octopod-shaped nanoparticles into a nanoparticle superlattice, although the existence of an ordered nucleus appears to be stable at the beginning, further growth does not follow monomer addition, and this is different from the monomeric addition assumption in CNT [36]. The existence of other competing factors during crystallization at the nanoscale may also lead to the growth of nanoparticle superlattices with self-limited size and shapes [37], which further points out the limitations of the CNT.

### 2.2. Data Processing Methods for Liquid-Phase Crystallization Observations

Image processing plays a critical role in the analysis of liquid-phase TEM data. In comparison to the standard image processing tools developed by optical microscopy [38], TEM images have much lower contrast in comparison to optical images and require a robust image processing algorithm to enhance the contrast of the images for high-throughput data analysis. The signal-to-noise ratio is also dependent on the electron beam dose rate, which is usually preferred to be low due to the potential electron beam damage and radiolysis effect [39,40]. Nanoparticles also include a huge library of shapes, which requires not only tracking the center positions of each particle but also their orientations and shapes [41]. The development of machine learning has significantly improved image processing speed and accuracy. In optical microscopy, counting and segmenting different cells has been demonstrated utilizing the image processing architecture of U-Net [42], and a similar neural network structure can also be applied to segment electron micrographs [43]. Because of the capability of simulating the electron micrographs based on materials properties and electron beam parameters, the manual labeling of a training dataset for machine learning can be greatly minimized, even avoided, as demonstrated by the successful tracking of nanoparticle etching and self-assembly kinetics in low-dose imaging conditions utilizing machine learning algorithms [43].

From the preprocessed images, the shape and location of each nanoparticle can be quantitatively determined. Depending on the different observations, follow-up analysis could be divided into several categories. For an analysis focusing on the morphological development of individual nanoparticles, the contour of the particle boundaries’ characterizations can be first extracted from the contrast-enhanced images, followed by measuring the directional center-to-edge distance and local curvature mapping during shape development, as well as general area evolution [44,45,46,47]. For an analysis focusing on the self-assembly process of a cluster of nanoparticles, tracking the center positions of each nanoparticle accurately is the first step, and local structural descriptions can be calculated from these spatial coordinates, including radial correlation functions, and order parameters for each specific symmetry of the crystalline lattice [48,49]. Based on the quasi-equilibrium thermodynamic assumptions, the anisotropic interparticle interactions between pairs of nanoparticles can also be extracted from liquid-phase TEM, allowing a unique method to probe the nanoscale interactions [50].

### 2.3. Computer Simulation for Crystallization Kinetics

Computer simulations are particularly important to the understanding of crystallization kinetics and for explaining the experimental observations. Common simulation methods for simulating crystallization kinetics include molecular dynamics, Monte Carlo simulation, and density functional theory [29,30,51]. In numerical simulations, the interparticle interactions can be tuned accurately, from the ideal hard-sphere to the Lennard–Jones potential, depending on the experimental conditions [5,52]. Going beyond testing the experimental observations and models, simulations can also study questions that are not possible to answer in real experiments. For example, the diffusion kinetics of nanoparticles can be affected by the long-range hydrodynamic interaction, which can be illustrated clearly by removing the hydrodynamic interaction between particles in computer simulations [53]. Similar methods also revealed that hydrodynamic interaction would shift the crystallization density in both passive and active colloidal systems [54]. Such testing of hydrodynamic interactions in computer simulations also rules out the explanation that a hydrodynamic interaction is the main reason for explaining the discrepancy of nucleation rates between theory and experimental observations [55].

Similar to the experimental system, with accurate spatial and orientational coordinates of each particle or atom, the phase transition kinetics can be accurately quantified. Local structural order parameters can be assigned to each particle, allowing the mapping of the symmetry transition pathway at the individual particle level [56]. For example, by distinguishing particles with a transient medium-range structural order in a supercooled liquid, it is found that the assumption of nucleation occurring from a homogenous disordered liquid might be oversimplified, and the temporal fluctuations in the structural order promote the nucleation events [57]. The development of machine learning tools, such as principal component analysis, diffusion maps, and support vector machines, also improves the throughput of computer simulation and reveals order parameters that are not commonly used based on structural symmetry descriptions [58,59,60].

## 3. Experimental Observations of Classical Nucleation Pathway

In CNT, individual particles will first come closer to form small embryos that share the same density and symmetry as the final products (Figure 2a). Before we discuss the experimental observations of the classical nucleation pathway, for clarity, we first briefly introduce the CNT and a few physical parameters that are critical to the quantitative classical nucleation model [1,61]. In the framework of the CNT, the system’s free energy (ΔG) is described by two terms: change of free energy proportional to the volume of the cluster ($\Delta G_{v}$) and increase in free energy proportional to the surface area ($\Delta G_{s}$). In three dimensions, assuming the cluster follows a spherical shape, the total free energy can be written as:(1)$$\Delta G=\Delta G_{v}+\Delta G_{s}=\frac{4}{3}\pi r^{3}\Delta g_{v}+4\pi r^{2}\gamma$$

Here, r is the radius of the nucleus; $\Delta g_{v}$ is the free energy change per volume; and γ is the free energy change when increasing the per surface area, which is better known as surface energy or surface tension. For crystallization, the final product usually has a lower free energy than the parent phase, resulting in $\Delta g_{v}<0$. Thus, Equation (1) predicts a critical nucleus size:(2)$$r_{\mathrm{crit}}=-\frac{2\gamma }{\Delta g_{v}}$$
at which ΔG reaches the maximum. In the CNT, cluster size smaller than $r_{\mathrm{crit}}$ will tend to dissolve instead of continuously growing into a large crystal.

**Figure 2 materials-16-02026-f002:**
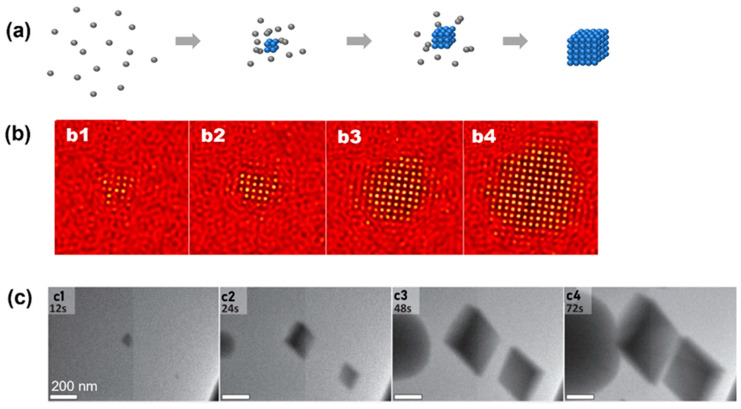
Simulation and experimental observations of the classical nucleation pathway: (**a**) Schematics illustrating the crystallization pathway following classical nucleation theory. (**b**) Simulation snapshots showing crystallization following classical nucleation theory. From (**b1**) to (**b4**), it shows the temporal evolution of crystal nuclei in computer simulation. Reproduced with permission from Ref. [35]. Copyright 2016 American Chemical Society. (**c**) Liquid-phase TEM snapshots of vaterite particles forming and directly nucleating from the solution following classical nucleation theory. From (**c1**) to (**c4**), the temporal evolution of the nuclei are captured by liquid-phase TEM. Reproduced with permission from Ref. [34]. Copyright 2014 American Association for the Advancement of Science.

The CNT has been observed in both numerical simulation and atomic crystallization by liquid-phase TEM. In a model phase-field crystal simulation in which the square lattice was the final stable lattice, the CNT has been observed by starting the crystallization close to the phase boundary of the square lattice [35]. As shown by the simulation snapshots in Figure 2b, a small cluster with square symmetry emerges from the liquid first, and this cluster propagates rapidly and grows into a square crystal. Liquid-phase TEM observations of the direct nucleation of calcite rhombohedra from solution are shown in Figure 2c [34]. It is noteworthy that although most particles in view followed the CNT, one particle also directly formed a (hemi-)spherical particle that, based on morphology, was either ACC or vaterite, and this suggests the concurrent direct formation of multiple phases from solution, which has also been predicted by computer simulations, where both classical and nonclassical pathways could occur in parallel [35].

Although the CNT has successfully explained the crystallization phenomenon, the model fails in several aspects. First, the crystallization rate is orders of magnitude different from what the CNT predicts [55], and this has raised new theories to explain crystallization, which is a nonequilibrium process [62]. Second, in terms of the crystallization kinetics, numerous experimental observations in atomic crystallization, nanoparticle, and micron-sized colloidal particle self-assembly have brought the conclusion that the CNT might have oversimplified the crystallization kinetics [2,4,6]. In the following section, several nonclassical crystallization pathways are discussed in detail from both observations in computer simulations and experimental observations.

## 4. Observations of Nonclassical Crystallization Pathways

### 4.1. Nonclassical Crystallization Pathway 1: Amorphous Structure below Critical Nucleus Size

One of the assumptions of the CNT is that the initial nuclei maintain the same density and symmetry during the whole growth process, which may not be true. Probing the nucleus at a subcritical size for an atomic system is challenging due to the subnanometer and short-living nature of these clusters, and previous work using a fluctuation transmission electron microscope suggests that the subcritical silver nuclei formed in a glassy solid have an amorphous nature [63]. As numerical simulation could accurately track the position of every single atom and reveal the cluster morphology of each atom during nucleation, it has been pointed out that in the initial stage, the subcritical clusters could appear as a mixture of different crystalline phases before the whole cluster transforms into a large postcritical crystalline lattice (Figure 3b) [64].

A similar pathway has been observed in liquid-phase TEM for an atomic crystallization system of a platinum nanocrystal [65]. Figure 3c shows the TEM images and corresponding fast Fourier transform (FFT) patterns at different times. The TEM images revealed that the initial precursors were tiny amorphous clusters, which was further confirmed by the diffusion FFT patterns. In this early nucleation state, the average sizes of the clusters were measured to be 1.0, 1.8, and 2.2 nm, indicating a general size limit among this nucleus. In Figure 3(c5–c8), the lattice fringes gradually became clear together with the diffraction spots corresponding to the (111) and (200) directions of the platinum lattice, and the average size of the clusters reached 2.0 nm. Interestingly, most of the clusters remained similar in size and shape compared with their amorphous counterparts, although the crystallinity of the nanoparticle changed completely. This shift of crystallinity at the initial stage of the nucleation is not predicted in the CNT. Similar observations have also been reported in the heterogenous nucleation of iron nanocrystals inside carbon nanotubes, where amorphous clusters were observed below the critical nucleus sizes (100–1000 atoms) [66].

### 4.2. Nonclassical Crystallization Pathway 2: Nucleation of Crystalline Phase from an Amorphous Intermediate

The nonclassical crystallization pathway involving the formation of a large amorphous cluster is one of the most observed nonclassical crystallization pathways in both simulations and experiments (Figure 4a) [15,29]. This crystallization pathway has been commonly referred to as a two-step crystallization pathway, in comparison to the one-step CNT [67], and a phenomenological model for this crystallization pathway has also been proposed [68]. Qualitatively, the amorphous cluster has a structure that is disordered but highly dense, sitting between the initial low-density disordered and final high-density ordered phases, thus lowering the interfacial energy and free energy barrier during nucleation compared with the CNT. This pathway can be accurately visualized via the computer simulation of a two-dimensional system (Figure 4b). From a large amorphous cluster, nuclei of a square symmetry started to emerge, and the whole cluster rapidly transformed into a square lattice. This nonclassical crystallization pathway has been reported in both atomic crystallization and nanoparticle superlattice formation processes, as discussed below.

For atomic systems, it is relatively challenging to image the amorphous structure in a solution with a single-atom resolution. An example is given in Figure 4c, where the crystallization of platinum nanoparticles started from an amorphous cluster, the amorphous nature of which could be confirmed by the fast Fourier transform pattern [65]. After imaging for approximately 80 s, a small platinum nucleus started to emerge from the center of the amorphous cluster, and the crystalline region expanded over time, with a diffraction pattern matching with the platinum lattice. A more challenging example of the crystallization of a metal–organic framework (MOF) nanocrystal shows the molecules first condensed in a cloudy solution-rich region, and the dense aggregates transformed into the final crystalline cubic nanoparticle (Figure 4d) [33]. Due to the difficulty of visualizing individual atoms in liquid-phase TEM, especially light atoms such as carbon (discussed in detail in Section 6), cryo-EM and molecular dynamic simulation were used to support the observations in liquid-phase TEM. An earlier study on gold nanoparticle nucleation has also reported a similar observation, involving a spinodal decomposition of the solution into regions with different concentrations of solute [69].

For nanoparticle systems, liquid-phase TEM can track the motion of individual particles accurately, and single-particle-based local structure analysis (discussed in detail in Section 2) is allowed to fully illustrate the crystallization kinetics. In an example of the assembly of gold nanospheres into a two-dimensional superlattice, the nanoparticles were found to first form into clusters, and the large cluster included multiple domains of different crystalline orientations (Figure 4e) [70]. The different domains inside the cluster then rotated and rearranged before achieving an extended ordered nanoparticle superlattice. In another three-dimensional example, the triangular nanoparticles first assembled into individual columnar structures, and the columns first formed a large-scale disordered cluster [49]. This amorphous liquid-like state lasted for over a hundred seconds before a stable solid nucleus existed inside the amorphous cluster. By visualizing the transition pathway on a two-dimensional density structure histogram, a clear transition was visualized, where the systems first evolved towards a state with a high-density but low structure order, and the crystalline domain with a high-density high-structure order appeared afterward.

It is also worth mentioning that this two-step crystallization pathway has also been reported in micron-sized colloidal systems [71]. A great advantage of micron-sized colloidal systems is that the interparticle interaction can be quantitatively modulated due to the well-understood interaction between individual colloids, and in situ experimental observations can be easily carried out with optical microscopy. In one example utilizing temperature-dependent depletion attraction to control the crystallization kinetics, different crystallization kinetics could be achieved in the same experimental system under different temperatures [67]. Interestingly, from the fluctuation of the clusters observed during the crystallization, the free energy barrier can be quantitatively measured to extract the surface energy and chemical potential, illustrating the lower free energy for a two-step crystallization pathway in comparison to the CNT.

### 4.3. Nonclassical Crystallization Pathway 3: Transition between Multiple Crystalline Structures before Achieving the Final Product

It is well known that in colloidal systems, a thermodynamically stable structure can have different symmetries depending on the temperature and density, even for the simple hard-sphere and Lennard–Jones model [52]. Before the system reaches its final state, intermediate states with a different crystalline symmetry could form first, and the lattice will then transform into the final lattice structure (Figure 5a). Previous experimental observations on micron-sized colloidal systems have suggested cross-symmetry pathways regardless of the final structure and interaction potential [52,72]. Such transitions between different lattice structures have also been observed in numerical simulations and atomic crystallization of calcium carbonate. In the example of the numerical simulation, the cluster converted into a metastable triangle nucleus instead of a final square nucleus in the first stage (Figure 5b) [35]. Then, the square phase formed on the surface of the triangle phase through heterogeneous nucleation, and all of the metastable phases transformed into the square phase. In the example of the experimental observation, calcium carbonate first formed into the aragonite phase, and a calcite rhombohedron then nucleated in apparent contact with the aragonitic bundle [34]. The calcite rhombohedron then grew continuously as the aragonitic bundle dissolved with a similar pathway to the simulation.

## 5. Crystal Growth Pathways after Nucleation Observed by Liquid-Phase TEM

After the initial nucleation and passing the critical nuclei size, the crystal will continue growing following the crystalline lattice symmetry. In the CNT, the growth of the cluster is via the monotonic attachment of single atoms or particles (Figure 6a). This growth can be observed in both the growth of single nanoparticles and nanoparticle superlattice. As shown in Figure 6b, under the hypersaturation condition, an individual nanoparticle grows and transforms its shape from a rhombic dodecahedron to tetrahedron, changing the surface facets from {110} to {111} under the illumination of the electron beam [73]. Such monomeric growth can also be observed during the expansion of a nanoparticle superlattice, and the surface profile steadily moves forward over time (Figure 6c), while the surface continuously fluctuates, suggesting a generic balance between thermal agitation and surface energy [74]. By analyzing the fluctuation spectrum of the interface between the superlattice and solution, the surface energy of the system can be quantitatively measured to explain the change of the exposed surface lattice orientation during the crystal growth [74].

Another common growth pathway is by coalescence between clusters, and the combined cluster will then relax to a shape close to a sphere to avoid the region with high surface curvature (Figure 6d). In Figure 6e, two individual gold nanoparticles came together to grow into a larger cluster, and the cluster gradually changed its shape into a dumbbell shape [75]. Different from the continuous increase in cluster size, the size of the cluster abruptly increased over a short period. Similar observations have also been made in nanoparticle superlattices. By electrostatically screening the interaction between gold nanoarrows, coalescence growth was observed between two large nanoparticle clusters (Figure 6f) [76]. Each large cluster had more than 100 nanoparticles, and when the two clusters became closer, the formation of a neck was observed, with some surface nanoparticles coming into physical contact first. The neck gradually expanded reducing the surface area with high surface curvature. The similarities in the growth behaviors between the atomic crystal and nanoparticle superlattice also point out the generic application of the crystallization theory over a broad range of materials models.

In the coalescence process, one special pathway is the oriented attachment of colloidal nanoparticles [77]. In this pathway, the colloidal particle will first dynamically move and readjust its orientation so that the crystalline lattices are aligned in their molecular structures even before the two particles physically join together. Liquid-phase TEM has been an ideal system to study such an orientated attachment process due to the fact of its capability to maintain the dynamic motion of nanoparticles in solution and track both the translational and rotational behaviors of the nanoparticles. In an example of studying the coalescence of gold nanoparticles in a mixture of sodium citrate solution with a 0.24 mM chloroauric acid (HAuCl4) solution, the oriented attachment between two gold nanospheres were tracked with an atomic lattice resolution [75]. It was found that the nanoparticles moved randomly when the separation between them was greater than twice the layer thickness of the adsorbed ligands, and the ligands guided the rotation of the particles in an aligned direction. The detailed interparticle interactions strongly affect the final morphology during the coalescence between nanoparticles, which has also been shown in another example of coalescing between two triangular nanoparticles [78]. In this example, the electrostatic repulsion between two nanoparticles can be quantitatively controlled, which changes the final structural morphology from linear chains to a continuous network. When the two particles joined together in solution, the specific bond angle was chosen based on the detailed morphology at the tip of the nanoparticles. Computer simulations of interparticle interactions have played an important role in illustrating the self-assembly kinetics in these systems [75,78]. Such oriented growth pathways have also been observed in natural systems, such as the formation of mesocrystals of iron oxides, and the special environment near the interface between existing crystals and solution plays an important role in driving the formation of these mesocrystals [79].

## 6. Discussions

It is noteworthy that not every crystallization pathway predicted in the simulation model was observed experimentally in the liquid-phase TEM studies based on our studies. For example, we did not find examples of nanoparticle systems that strictly follow the classical nucleation pathway. This might due to the difference in the interaction potential between the nanoparticle system and numerical simulations [80]. Interestingly, previous studies on micron-sized colloidal systems have been well understood from computer simulation, [5] and the transition from classical to nonclassical crystallization kinetics has been achieved in a single experimental system with tunable interactions [67]. This further points out the importance of studying the crystallization kinetics for the rich library of colloidal nanoparticles, linking the relatively well-understood atomic and micron-sized colloidal systems.

As we introduced in Section 2, computer simulations play a critical role in understanding crystallization dynamics and kinetics [81]. The nonclassical crystallization kinetics involving a large amorphous cluster (path 2 in Figure 1) was first reported in 1997 via computer simulation of a model protein system [30], and it was pointed out that through this nonclassical pathway, the protein crystallization speed can be accelerated. Such a nonclassical crystallization pathway can be used to help the drug formation process in industrial production. Computer simulations have also played a critical role in explaining and verifying the experimental observations from atomic structure transitions to nanoparticle superlattice dynamics [49,82] due to the limitations of contrast and resolution from experimental TEM observations. Recent progress in machine learning and data processing has provided novel paradigms to study self-assembly and structural transitions on a large scale, offering opportunities to illustrate long-lasting problems in soft materials and biological systems, which are discussed in detail in the following paragraphs [43,58,83].

Although liquid-phase TEM has achieved great success over the past decades, there are several crucial challenges faced by this technique that need to be carefully considered for each observation [21,84]. We summarize several important aspects below, together with potential strategies to overcome these limitations:●First, the electron beam will interact with the materials, as well as the water molecules, during illumination, creating ions and radicals that affect the aqueous environment. Previous work on numerical simulation carefully illustrated all of the radiation kinetic processes when an electron beam interacts with water and predicted the temporal evolution of different species in an aqueous environment [39]. Such radiolysis has also been independently verified in the observations of nanobubble formation and growth inside liquid-phase TEM [85,86]. These radiolysis species become crucially important for the study of the crystallization of organic or biological materials, such as the assembly of peptides [87] and virus shells from proteins [88]. Although radiolysis damage was still observed during the imaging process of liquid-phase TEM, some biological samples were reported to be approximately ten times more stable than frozen-hydrated states, which are used in cryo-EM [89]. In one example, the dynamic motion of adeno-associated virus (AAV) has been achieved after optimizing the liquid thickness and electron beam dose rate, and the resolution was comparable to cryo-EM [89]. These damages can be further improved by the technical advancements of electron microscopy, especially in the imaging routine and detector efficiencies;●Second, the intensity contrast of TEM comes from the intrinsic properties of individual elements, and it is challenging to image biological samples [90,91,92]. One of the reasons that previous studies chose materials with relatively large element numbers, such as gold and platinum, is due to the fact of their strong contrast under electron beam illumination. The contrast of materials under electron beam illumination can be quantitatively evaluated by Beer’s law, where the transmitted intensity of the electron beam decays exponentially with the increasing thickness of the materials [43]. Table 1 includes the mean free path of an electron beam in materials based on the database released by the National Institute of Standards and Technology (NIST) [93]. At typical accelerating voltages used for TEM imaging, the mean free path of noble metals is approximately a third of carbon, which is the main component of all biological materials. This challenge is not only limited to liquid-phase TEM, several contrast enhancement techniques have been developed in conventional TEM to improve the contrast of elements, including phase contrasting methods [94] and chemical staining techniques [95];●Third, the liquid chamber for liquid-phase TEM is relatively small, and a typical thickness is several hundred nanometers, which is different from the normal experimental conditions, and the substrate plays a critical role in the crystallization kinetics [49]. The small volume of the chamber could affect the crystallization kinetics from multiple different aspects. On the one hand, it has been pointed out in both experiments and simulations that the crystallization kinetics for systems with different size limits are different [96,97]. On the other hand, the physical existence of boundaries in liquid-phase TEM will also affect the crystallization kinetics by attracting the atoms or nanoparticles to the vicinity of the membrane [70,78]. This boundary might also align the nanoparticles into a certain orientation, facilitating the self-assembly process [49]. Lastly, this macroscopically uniform chamber film might not be nanoscopically uniform in terms of surface chemistry, as pointed out by similar spatial mappings between the heterogeneous nucleation domains imaged by liquid-phase TEM and the functional group domains mapped by liquid-phase AFM [98];●Forth, TEM collects all of the electrons transmitting through the sample, which includes all materials and beam trajectories, making it difficult to interpret the three-dimensional structural details. This challenge applies to all imaging techniques relying on the transmission of the probe through the sample, such as the conventional optical microscope. To overcome this challenge, a confocal microscope introduces a pinhole to reduce the light collected from the region out of the focal plane, and this would improve the resolution along the beam direction [99]. With such improved resolution, it is then possible to allow the scanning of the focal plane through the sample and the differentiation of the images captured at different depths. Similar techniques have also been applied to electron microscopes to capture “depth sectioning” images, and the three-dimensional reconstruction of nanostructures based on sectioning images has been also demonstrated [100]. Another strategy is to compare the contrast of experimental TEM with simulation models, as they can be quantitatively simulated after taking all the materials along the electron beam and noise origins into consideration [43]. This could potentially be used to illustrate the three-dimensional organization of materials inside the liquid-phase chamber [43,101]. For example, in a colloidal superlattice where multiple layers of nanoparticles stack together, understanding the particle locations in each layer becomes crucial to understand the two-dimensional projections captured by liquid-phase TEM. In a study of the assembly of gold nanospheres at the solid–liquid interface, it is reported that the second layer did not necessarily follow the close-packed hexagonal lattice, following the first layer [102]. Instead, it could pack into a quasicrystalline lattice if the dielectric constant of the solution could be modulated correctly. Such mechanism studies on the formation of multilayer structures offer important insights into the growth of three-dimensional materials, such as the nanoparticle substrates used in novel devices [103].

The advancements in the technology of both electron microscopes, detectors, and data processing pipelines will all facilitate overcoming these challenges [104]. For example, it has been noted that scanning through all locations during the scanning transmission electron microscopy might not be necessary for the atomic reconstruction of nanoparticles [105]. By reducing the scanning sites during imaging, the acquisition time can be significantly improved, allowing the observation of fast dynamic processes. The reduced acquisition time also helps minimize the electron beam damage to the soft materials. Another method to reduce electron beam damage on soft materials is by adding radical scavengers such as 2-proponal, which has been demonstrated in the imaging of thermal responsive poly(N-isopropylacrylamide) microgels [106]. By utilizing this strategy, successful characterizations of the thermoresponsive behaviors of these microgels are achieved using liquid-phase TEM, illustrating the detailed volume shrinkage dynamics. However, the microgels shrinkage dynamics were irreversible, probably due to the electron beam damage or electron beam-induced water radiolysis products. All these limitations might be resolved in the future with the development of efficient electron detectors and optimized imaging protocols.

## 7. Conclusions

Liquid-phase TEM has offered its unique strength in understanding crystallization kinetics, which can be used to explore a variety of different systems ranging from nanomaterials synthesis and electrochemical reactions to virus shell formation and biomineralization. Advancements in both the electron microscope and computer simulations will significantly contribute to the understanding of the crystallization kinetics in different systems. Combed with other in situ and ex situ techniques, liquid-phase TEM will continuously offer observations of crystallization kinetics at the nanoscale, providing new insights into the fundamental understanding of the nanoscale interactions and offering novel design rules for functional nanomaterials. A detailed comparison between simulation and experimental observations of crystallization pathways, such as the four pathways summarized in this review, will help the construction of materials with the desired surface and bulk properties. With a single-particle resolution, the direct mapping of the anisotropic interparticle interactions between pairs of nanoparticles can be extracted from experiments to compare with theory [43,50], and the growth of nanostructures with pre-engineered sizes and shapes can be further refined based on the understanding of the kinetic crystallization pathways.

## Figures and Tables

**Figure 1 materials-16-02026-f001:**
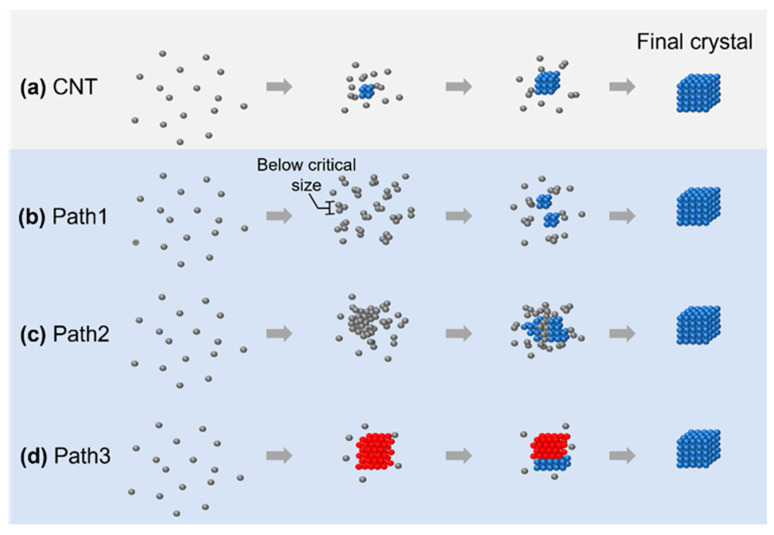
The four different crystallization pathways overviewed: (**a**) crystallization pathway described by classical nucleation theory; (**b**–**d**) three nonclassical nucleation pathways: formation of an amorphous cluster below critical nucleus size (**b**), nucleation of the crystalline phase from an amorphous intermediate (**c**), and transition between multiple crystalline structures before achieving the final product (**d**). The gray/red/blue spheres represent atoms or particles in a disordered/metastable crystalline/stable final crystalline structure, respectively.

**Figure 3 materials-16-02026-f003:**
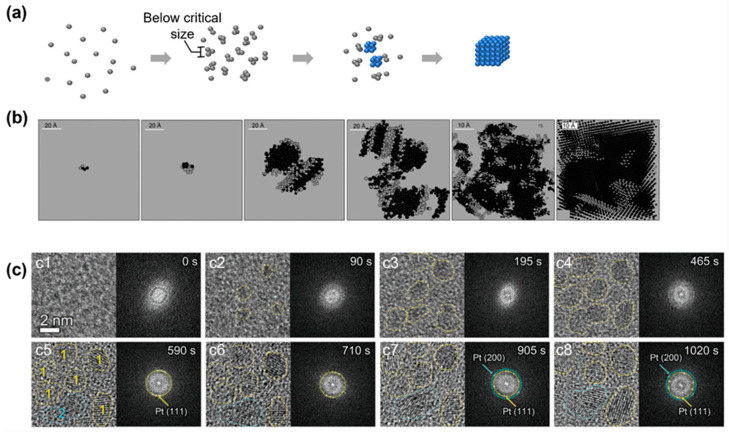
Simulation and experimental observations of the nonclassical crystallization pathway with the formation of an amorphous cluster below the critical nucleus size: (**a**) Schematics illustrating the nonclassical crystallization pathway and highlighting the amorphous structure below the critical size. (**b**) Simulation snapshots of clusters of silver atoms at different crystallization conditions to illustrate the amorphous structure below the critical nucleus size. From left to right, the systems were cooled down with cooling rates of 6 × 10^13^, 5 × 10^13^, 10^13^, 5 × 10^12^, 2 × 10^12^, and 10^12^ K/s, respectively. Reproduced with permission from Ref. [64]. Copyright 2010 Science China Press and Springer-Verlag Berlin Heidelberg. (**c**) Liquid-phase TEM observations of the nucleation and growth of platinum nanocrystals that highlight the amorphous structure below the critical size. From (**c1**) to (**c8**), the temporal evolution of the nuclei is captured by liquid-phase TEM and corresponding fast Fourier transform of each image is shown on the right. Reproduced with permission from Ref. [65]. Copyright 2022 Wiley Materials.

**Figure 4 materials-16-02026-f004:**
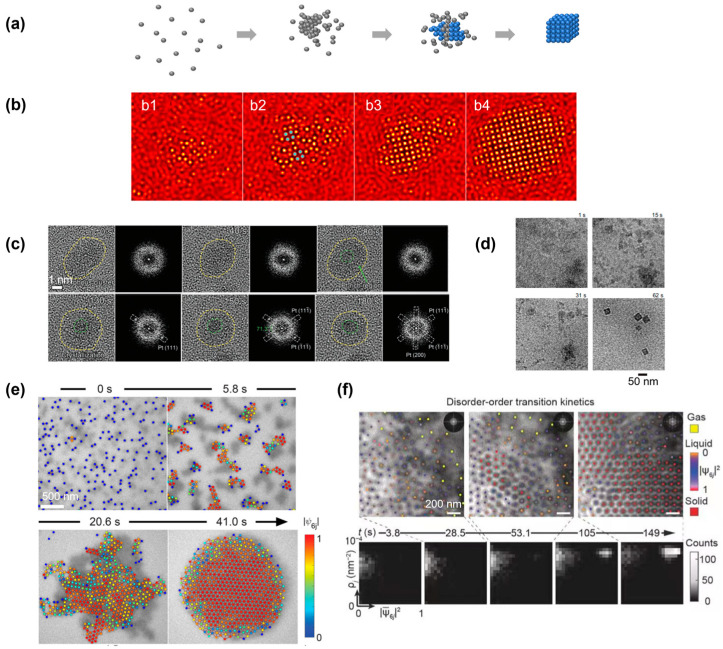
Simulation and experimental observations of the nonclassical crystallization pathway in which the crystalline domain nucleated from a large amorphous precursor: (**a**) Schematics illustrating the nonclassical crystallization pathway and highlighting the formation of an amorphous precursor first, from which the crystalline domain nucleated. (**b**) Simulation snapshots illustrating the formation of a large amorphous precursor first, and the crystalline domain (highlighted in blue) nucleated from inside of the cluster. From (**b1**) to (**b4**), the temporal evolution of a single nucleus is shown in computer simulation. Reproduced with permission from Ref. [35]. Copyright 2016 American Chemical Society. (**c**) Liquid-phase TEM observations of the structural evolution of a large amorphous cluster of Pt and the transformation to a crystalline structure. Fast Fourier transformations are also included to highlight the structure transition. Reproduced with permission from Ref. [65]. Copyright 2022 Wiley Materials. (**d**) Liquid-phase TEM images showing the crystallization process of a ZIF-8 metal–organic framework nanocube from solution. Reproduced with permission from Ref. [33]. Copyright 2021 National Academy of Science. (**e**) Liquid-phase TEM images showing the two-step crystallization of gold nanospheres. Reproduced with permission from Ref. [70]. Copyright 2022 American Chemical Society. (**f**) Time-lapse liquid-phase TEM images showing the real-time crystallization process of a high concentration of triangular-shaped gold nanoparticles. The density structure histograms at the bottom highlight the formation of an amorphous structure first before the transition to the final crystalline lattice. Corresponding order density histograms. Reproduced with permission from Ref. [49]. Copyright 2019 Springer Nature.

**Figure 5 materials-16-02026-f005:**
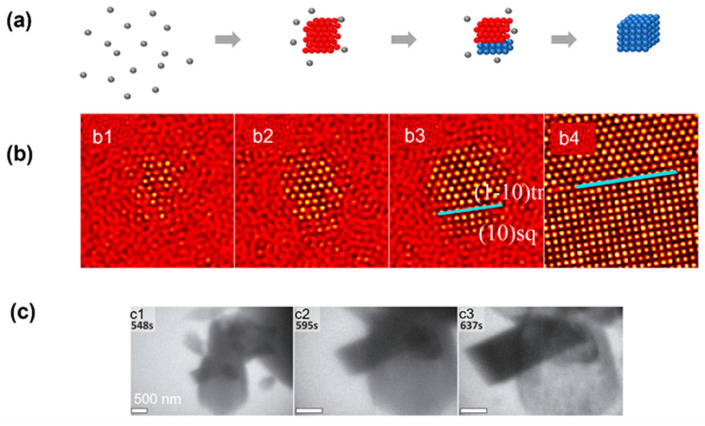
Simulation and experimental observations of the nonclassical crystallization pathway with the transition between multiple crystalline structures: (**a**) Schematics illustrating the nonclassical crystallization pathway, highlighting the formation of an intermediate crystalline lattice before achieving the final product. (**b**) Simulation snapshots illustrating the formation of an amorphous intermediate triangular lattice first and the transition into a square lattice through heterogeneous nucleation. From (**b1**) to (**b4**), the temporal evolution of a single nucleus is shown, and the blue line indicates the boundary between two phases. Reproduced with permission from Ref. [35]. Copyright 2016 American Chemical Society. (**c**) Liquid-phase TEM observations of the heterogeneous nucleation of calcite crystal occurring on aragonite, followed by calcite growth and concomitant dissolution of the aragonitic bundle. From (**c1**) to (**c3**), the temporal evolution of the nucleation of calcite crystal on the surface of aragonite is captured by liquid-phase TEM. Reproduced with permission from Ref. [34]. Copyright 2014 American Association for the Advancement of Science.

**Figure 6 materials-16-02026-f006:**
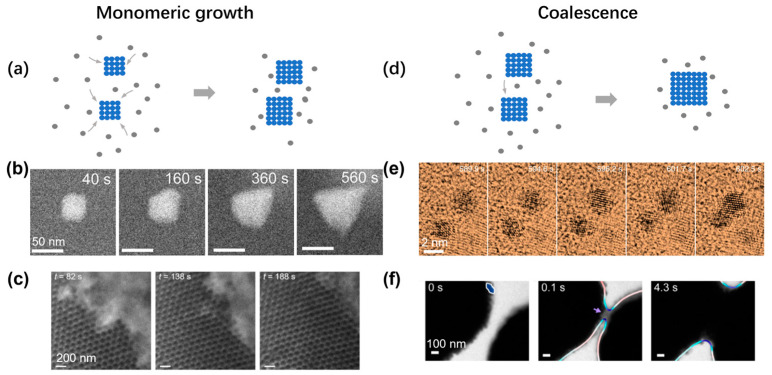
Experimental observations of crystal growth inside liquid-phase TEM: (**a**) Schematic illustrating the growth of crystal via monomeric attachment. (**b**) Time series of liquid-phase TEM showing the gradual growth of an individual nanoparticle. Reproduced with permission from Ref. [73]. Copyright 2021 American Chemical Society. (**c**) Time series of liquid-phase TEM showing the growth of an extended nanoparticle superlattice. Reproduced with permission from Ref. [74]. Copyright 2020 Springer Nature. (**d**) Schematic illustrating the growth of crystal via coalescence between two particles. (**e**) Time series of liquid-phase TEM showing the coalescence between two individual nanoparticles. Reproduced with permission from Ref. [75]. Copyright 2018 Springer Nature. (**f**) Time-lapse liquid-phase TEM images showing the coalescence of clusters assembled from gold nanoarrows. Reproduced with permission from Ref. [76]. Copyright 2020 American Chemical Society.

**Table 1 materials-16-02026-t001:** Inelastic mean free path of different elements and energy from the database. The data were extracted from Ref. [93].

Element	Element Number	Energy (eV)	Mean Free Path (nm)
Carbon	6	1000	3.270
		2000	5.297
Phosphorous	15	1000	2.481
		2000	4.329
Sulfur	16	1000	2.399
		2000	4.175
Silver	47	1000	1.125
		2000	1.827
		3000	2.425
Platinum	78	1000	0.972
		2000	1.382
Gold	79	1000	1.378
		2000	2.340
		3000	3.204

## Data Availability

This study does not generate new data.

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
