# Peer review of "Direct Imaging of the Kinetic Crystallization Pathway: Simulation and Liquid-Phase Transmission Electron Microscopy Observations"

_materials, 2023, doi:10.3390/ma16052026_

Round 1

Reviewer 1 Report

The authors submitted a review manuscript entitled “Direct Imaging of the Kinetic Crystallization Pathway: Simulation and Liquid-Phase Transmission Electron Microscopy Observations”.  The authors discussed several crystallization pathways captured by the liquid-phase transmission electron microscopy technique and compared the observations with computer simulation. The authors discuss the computer simulation in developing an understanding of the crystallization pathways and show the future perspectives on the investigation at the nanoscale. The authors refer to appropriate references and the drawn background on the topic seems acceptable. The combination of simulation and experimental techniques allows for a detailed understanding of the kinetic pathway of crystallization, which is crucial for materials engineering. Overall, the article provides a valuable contribution to the field of materials science.

 The manuscript requires some minor revisions:

To enhance the readability of the article, the equations should be listed separately and appropriately numbered, e.g. line 239, 243.

 Figure 3: You should use integer powers in specifying the cooling rates.

The authors claim that liquid-phase TEM will offer observations of crystallization kinetics at the nanoscale providing new insights into the fundamental understating of the nanoscale interactions and offering novel design rules for functional nanomaterials.
These potential novel design rules should be briefly discussed.

Author Response

Response to reviewer 1.

We greatly appreciate the reviewer’s comment that our review is “a valuable contribution to the field of materials science.” Below are detailed responses to the reviewer’s comments.

Comment 1: To enhance the readability of the article, the equations should be listed separately and appropriately numbered, e.g. line 239, 243.

Response: Thanks for the suggestions, and we have separated the equations and numbered them.

Comment 2: Figure 3: You should use integer powers in specifying the cooling rates.

Response: Thanks for pointing this out, and we have updated the values using integer powers in the caption of Figure 3.

Comment 3: The authors claim that liquid-phase TEM will offer observations of crystallization kinetics at the nanoscale providing new insights into the fundamental understating of the nanoscale interactions and offering novel design rules for functional nanomaterials. These potential novel design rules should be briefly discussed.

Response: Thanks for the valuable suggestions. We have added the following discussion to the end of our conclusions.

“The detailed comparison between simulation and experimental observations of crystallization pathways, such as the four pathways summarized in this review, will help the construction of materials with the desired surface and bulk properties. With single particle resolution, direct mapping of the anisotropic interparticle interactions between pairs of nanoparticles can be extracted from experiments to compare with theory,[43,50] and the growth of nanostructures with pre-engineered size and shapes can be further refined based on the understanding of kinetic crystallization pathways.”

Reviewer 2 Report

Ref_comments to the paper titled as “Direct Imaging of the Kinetic Crystallization Pathway: Simulation and Liquid-Phase Transmission Electron Microscopy Observations” written by the authors: Zhangying Xu and Zihao Ou.

Despite of a large number of works in the crystallization process evidences this

Important process of crystallization in materials science occupies an essential place and requires further study, taking into account both new knowledge and new equipment. From this point of view the current article is actual and modern.

For the first, the authors have made nice literature search connected with the analysis of 104 references. Good! The manuscripts published last 5 years have been included in this consideration as well.

This paper has so interesting illustrations for the four types of different crystallization pathways, which can predict the difference in the kinetic process.

These paths show 4 options that can be used in the construction of new composites, including modification of the volume of the material, and the surface of the material. Nice! Good prediction supported by the calculations.

But, in this concern, I would like to ask the authors about the followings:

1). Why they discuss in detail namely the CNTs applications? So many other nanostructures used in the materials construction are now considered: graphene, graphene oxides, quantum dots, Januce NPs, etc. Please give the unique features of the CNTs, which you have used for the discussion.

2). Why you have not used the order parameters in your consideration? It is connected with the refractive properties change, which are important to modify all basic characteristics of the structured materials. Have the authors the ellipsometry obtained data for all types of the discussed kinetic crystallization pathway?

The materials of the current paper are not in contradiction with our basic physical knowledge.

The Conclusion part are accumulated the important analysis made in this article, but it should be extended according to 4 ways of the discussed process.

As for my local opinion, this paper can be published after minor correction.

Author Response

Response to reviewer 2.

We appreciate the reviewer’s positive comments from the reviewer. The reviewer thinks that our review is “the current article is actual and modern.” The reviewer also points out that the four pathways we summarized are valuable, saying “these paths show 4 options that can be used in the construction of new composites, including modification of the volume of the material, and the surface of the material.” Apart from these, the reviewer give us some suggestions, and below are detailed responses to the reviewer’s comments.

Comment 1: Why they discuss in detail namely the CNTs applications? So many other nanostructures used in the materials construction are now considered: graphene, graphene oxides, quantum dots, Januce NPs, etc. Please give the unique features of the CNTs, which you have used for the discussion.

Response: We are very sorry about the misunderstanding. Here CNT means classical nucleation theory, instead of carbon nanotube as the reviewer points out. To avoid this confusion, we have changed or removed all the “CNT” in the figure captions to avoid potential misunderstanding. Apart from that, we also included an example showing how the crystallization of 2D materials can be studied in liquid-phase TEM in line 126.

Comment 2: Why you have not used the order parameters in your consideration? It is connected with the refractive properties change, which are important to modify all basic characteristics of the structured materials. Have the authors the ellipsometry obtained data for all types of the discussed kinetic crystallization pathway?

Response: Thanks for pointing out that we have missed this part of discussions. Following the reviewer’s suggestions, we looked into that and find literature using other order parameters. Due to the focus of our review, we didn’t extensively dive into this direction as we focused more on real-space structure imaging technique. We have added the following discussion:

“Crystallization process can also be studied in-situ by monitoring order parameters that are not directly related to the structure of the materials, such as ellipsometry[27] and uv-vis spectroscopy.[28]”

Comment 3: The Conclusion part are accumulated the important analysis made in this article, but it should be extended according to 4 ways of the discussed process.

Response: Thanks for the valuable suggestions. We have added the following discussion to the end of our conclusions.

“The detailed comparison between simulation and experimental observations of crystallization pathways, such as the four pathways summarized in this review, will help the construction of materials with the desired surface and bulk properties. With single particle resolution, direct mapping of the anisotropic interparticle interactions between pairs of nanoparticles can be extracted from experiments to compare with theory,[43,50] and the growth of nanostructures with pre-engineered size and shapes can be further refined based on the understanding of kinetic crystallization pathways.”

Reviewer 3 Report

In manuscript materials-2222572, the authors have a review on the use of liquid-phase transmission electron microscopy (TEM) to observe crystalization pathways. The authors present classical nucleation theory and some of the issues associated with its assumptions. They also present alternative pathways to crystallizations and examples of these pathways being observed through liquid-phase TEM. Finally, the authors present limitations to the TEM technique in the context of nanomaterial nucleation. This is a well-written work that covers the appropriate literature.

Conclusion: Provided the authors can address the minor comments below, this manuscript may be published in Materials without further review.

Minor comments:

  1. On line 26 in the abstract, “...developing a mechanistic to understand…”  I believe the authors are either missing the word “approach” after “mechanistic” or they meant to say mechanism.

  2. Reference 71 seems to be incomplete. Please double-check its details.

Author Response

Response to reviewer 3.

We greatly appreciate the suggestions from the reviewer and thanks for the positive feedback from the reviewer. Here are the changes we made based on the reviewer’s suggestions.

Comment 1: On line 26 in the abstract, “...developing a mechanistic to understand…”  I believe the authors are either missing the word “approach” after “mechanistic” or they meant to say mechanism.

Response: Thanks for pointing out this mistake. We have added the word “approach” following the suggestion.

Comment 2: Reference 71 seems to be incomplete. Please double-check its details.

Response: Thanks for pointing out this important reference issue. The original reference 71 was published on arXiv as a preprint and we have replaced it with two new references (Ref. 73 and Ref. 75) for better illustration.